# Identifying priority double-duty actions to tackle the double burden of malnutrition in infants and young children in Peru: Assessment and prioritisation of government actions by national experts

**Violeta Magdalena Rojas Huayta**[1]*, **Rebecca Pradeilles**[2,3], **Hilary M. Creed-Kanashiro**[4], **Emily Rousham**[2], **Doris Delgado**[1], **Rossina Pareja**[4], **Edwige Landais**[3], **Nervo Verdezoto**[5], **Emma Haycraft**[2], **Michelle Holdsworth**[3]

1 Facultad de Medicina, Departamento de Nutrición, Universidad Nacional Mayor de San Marcos (UNMSM), Lima, Perú, 2 Centre for Global Health and Human Development, School of Sport, Exercise and Health Sciences, Loughborough University, Loughborough, United Kingdom, 3 UMR MoISA (Montpellier Interdisciplinary Centre on Sustainable Agri-Food Systems), (Univ Montpellier, CIRAD, CIHEAM-IAMM, INRAE, Institut Agro, IRD), Montpellier, France, 4 Instituto de Investigación Nutricional, Lima, Perú, 5 School of Computer Science and Informatics, Cardiff University, Cardiff, United Kingdom

* vrojashu@unmsm.edu.pe

## Abstract

Multiple forms of malnutrition coexist in infants and young children (IYC) in Peru. The World Health Organization has proposed double-duty actions (DDAs) to simultaneously address undernutrition and overweight/obesity. We assessed current implementation of- and priority for- government-level actions to tackle multiple forms of malnutrition in IYC in Peru. Mapping of current policy activity was undertaken against 47 indicators of good practice for five DDAs (exclusive breastfeeding, complementary feeding, food marketing, maternal nutrition, pre-school nutrition; assessed by 27 indicators) and for the enabling policy environment, i.e., 'infrastructure support' (health in all policies, platforms for interactions, financing, monitoring, governance, leadership; assessed by 20 indicators). Interviews with 16 national experts explored views on the level of and barriers to implementation of DDAs and infrastructure support, as well as their prioritisation based on likely impact and feasibility. The level of implementation of actions was categorised into two groups (agenda setting/formulation vs. implementation/evaluation). Mean scores were generated for prioritisation of DDAs and infrastructure support. Deductive qualitative analysis was undertaken to identify barriers that influence policy implementation. Only 5/27 DDA indicators were reported as fully implemented by all national experts (*international code that regulates the marketing of breastmilk substitutes, iron supplementation for IYC, micronutrient powders in IYC, iron/folic acid supplementation in pregnant women, paid maternity leave*). Only 1/20 infrastructure support indicator (*access to nutrition information*) was rated as fully implemented by all experts. Barriers to implementing DDAs and infrastructure support included: legal feasibility or lack of regulations, inadequate monitoring/evaluation to ensure enforcement, commercial influences on policymakers, insufficient resources, shifting public health priorities with the

**Data Availability Statement:** All relevant data are within the paper and its Supporting Information files.

**Funding:** Funding: This study was supported by the UK Medical Research Council (MR/S024921/1) and CONCYTEC/FONDECYT Perú (032-2019) through the Newton Fund. The funders had no role in study design, data collection and analysis, decision to publish, or preparation of the manuscript

**Competing interests:** The authors have declared that no competing interests exist.

COVID-19 pandemic and political instability. The experts prioritised 12 indicators across all five DDAs and eight infrastructure support indicators. Experts highlighted the need to improve implementation of all DDAs and identified ways to strengthen the enabling policy environment.

## Introduction

The double burden of malnutrition (DBM) is defined as the coexistence of malnutrition due to deficiency (i.e., micronutrient deficiencies, underweight, childhood stunting and wasting) and excess (i.e., overweight/obesity and diet-related non-communicable diseases- DR-NCDs), which can affect countries, households, and individuals [1]. Both undernutrition and overweight/obesity have, for a long time, been addressed as separate public health problems requiring different sets of solutions. However, the current nutrition reality shows that they are interconnected, hence requiring interventions to be reshaped to address multiple forms of malnutrition simultaneously [2].

To address the DBM, so called 'double-duty actions' (DDAs), i.e., interventions, programmes, and policies with the potential to simultaneously reduce the risk or burden of undernutrition and overnutrition, have been proposed by the World Health Organization (WHO) [3]. These include exclusive breastfeeding in the first 6 months and continued breastfeeding up to 2 years; adequate complementary feeding in IYC from 6 months; maternal nutrition and prenatal care programmes; school feeding policies and programmes; and marketing regulations. These are not necessarily new actions, as they include those already used to address individual forms of malnutrition but have the potential to become DDAs that provide integrated strategies for multiple forms of malnutrition [2–4].

Globally, the most prevalent forms of malnutrition in children under five are stunting and anaemia. It is estimated that nearly a quarter of all children under five are stunted [5], 39.8% of children aged 6–59 months suffer from anaemia [6], and childhood overweight and obesity is increasing rapidly in almost all countries, with no sign of slowing down [5]. In Peru, the nutrition landscape has evolved over recent years. There has been some positive change. Firstly, the prevalence of stunting in children under five has decreased from 23.2% in 2010 to 11.5% in 2020 [7]. Secondly, anaemia in children under three has decreased from 43.5% (2015) to 38.8% (2021) but remains high and is currently considered one of the main public health nutrition problems in Peru [7]. The prevalence of overweight/obesity in children under five was estimated at 9.6% in 2021 [7]. However, overweight/obesity in children aged 5–9 years is considerably higher (37.4% in 2017–2018) [8] which could indicate that the increase in prevalence occurs from early childhood.

In Peru, the generic government policy aimed at improving the nutritional status of the population is the *National Multisectoral Health Policy 2030*, which includes a priority objective of the promotion of healthy behaviours like healthy eating [9]. This multisectoral policy is aligned with the *Plan for the reduction and control of maternal and child anaemia and chronic child undernutrition* [10] and with governmental actions aimed at reducing overweight and obesity in children and adolescents, such as the *Law for the promotion of healthy eating for children and adolescents* [11]. The objectives declared in the *Early Childhood Development Results-Oriented Budget Programme* are the reduction to 6% (chronic undernutrition), 38% (anaemia) and 5% (overweight/obesity) in children under five [12]. Policies and declared objectives regarding the DBM in Peru have not yet been implemented, therefore it is necessary to reorient public policies to address the DBM.

In this context, this study sought to respond to the need for evidence that exists in Peru to assess the current implementation of- and priority for- government-level actions to tackle multiple forms of malnutrition in infants and young children (IYC).

## Materials and methods

### Study design

The methodological process was divided into three steps: i. mapping current policy actions; ii. assessing the level of implementation of government actions; and iii. prioritising government actions (**Fig 1**) during January 2020 to June 2021. The methods incorporated some of the approaches in the Healthy Food Environment Policy Index (Food-EPI); a tool developed to measure the extent of implementation of healthy food environment policies for preventing DR-NCDs [13,14]. This was adapted to assess the implementation of the five WHO-recommended DDAs [3]. For this study, we additionally used qualitative methods to explore the reasons behind experts' ratings on the level of implementation and prioritisation.

**Step 1: Mapping of current policy actions: Double-duty actions and 'infrastructure support'.** In step 1, the following three sub-steps were developed: designing a benchmarking tool for double-duty policy, mapping evidence for policy activity against the benchmarks; and validating the evidence with experts. The details of each sub-step are shown below.

### • Step 1.1: Designing a benchmarking tool for double-duty policy

A tool to assess the five WHO-recommended DDAs and elements of the enabling environment ('infrastructure support') was designed, using the same steps as in the Food-EPI tool but adapted for DDAs (Fig 2).

All five WHO DDAs were included (assessed by 27 indicators): exclusive breastfeeding, complementary feeding, food marketing, maternal nutrition, preschool nutrition (adapted from the school setting because our target group was IYC). The enabling policy environment, i.e. 'infrastructure support' was assessed with 20 indicators across six domains (health in all policies, platforms for interactions, financing, monitoring, governance, leadership). We adopted some of the indicators from the Food-EPI tool that were relevant for our outcomes (overweight/obesity, stunting and anaemia in IYC).

The selection of these indicators (**Fig 3** and **S1 Table**) involved synthesising evidence of what type of interventions and benchmarks should be included in each DDA from recent and major global reports and academic publications on recommended interventions, policies, and programmes to tackle all forms of malnutrition in IYC.

### • Step 1.2: Mapping evidence for policy activity against the benchmarks

A review of the evidence for current policy activity was conducted to assess the implementation of the 47 good practice indicators included in the tool (27 policy indicators and 20 infrastructure indicators). This yielded 164 documents. Evidence of government action in each of the policy and infrastructure support indicators, and across all 47 indicators, was systematically identified and collected via several steps: firstly, key public/government organisations involved in the different policy and infrastructure support indicators were identified, including organisations' websites. Government websites and those of other institutions (e.g., Food and Agriculture Organization (FAO), WHO, United Nations International Children's Emergency Fund (UNICEF)) were systematically searched for evidence of action. These websites were then searched for evidence on policies and/or infrastructure support. Evidence of Peruvian government policy action in the period 2015–2020, including those implemented prior to that

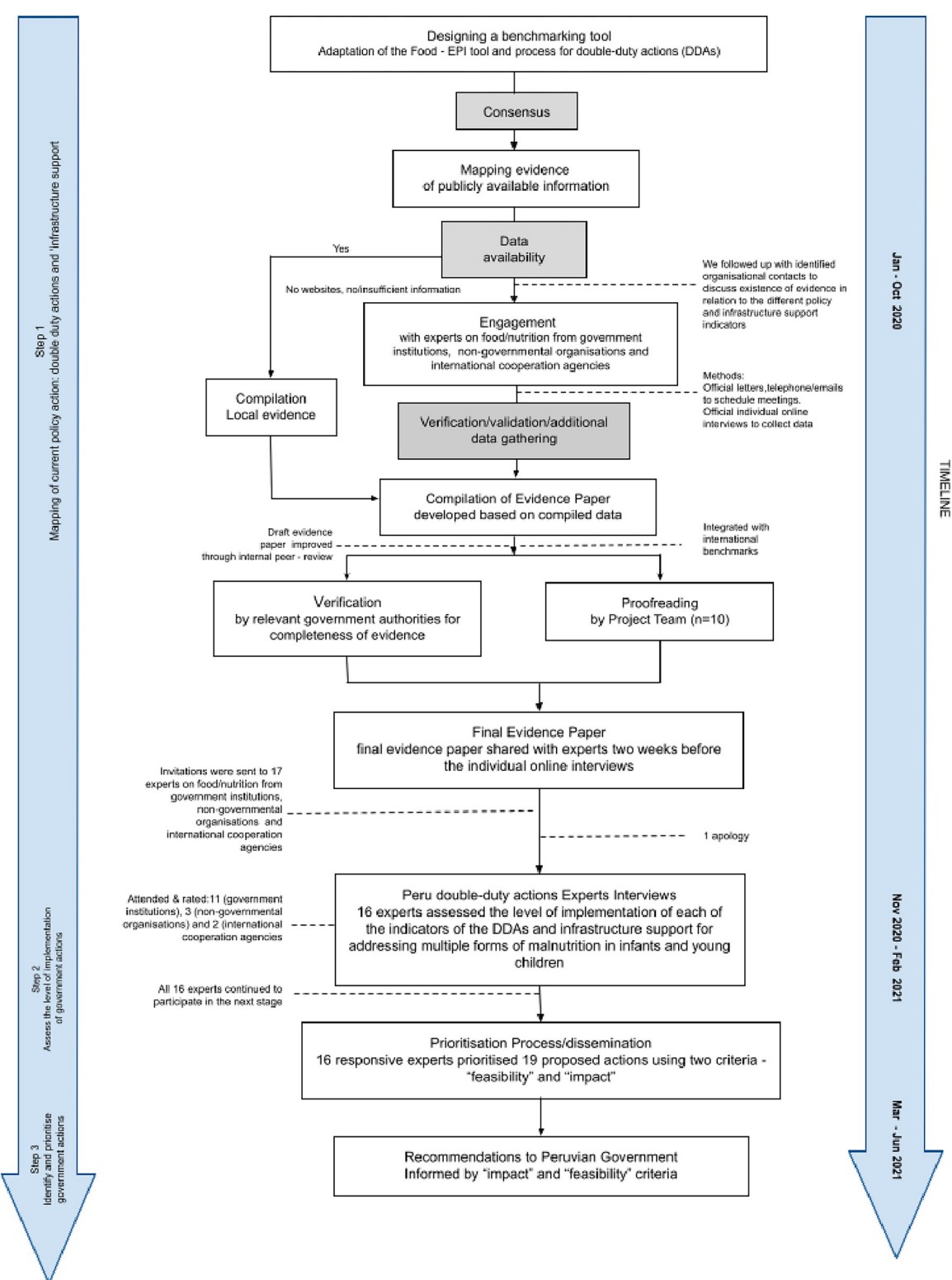

**Fig 1. Process for assessing the implementation and prioritisation of double-duty actions in Peru 2020–2021.**

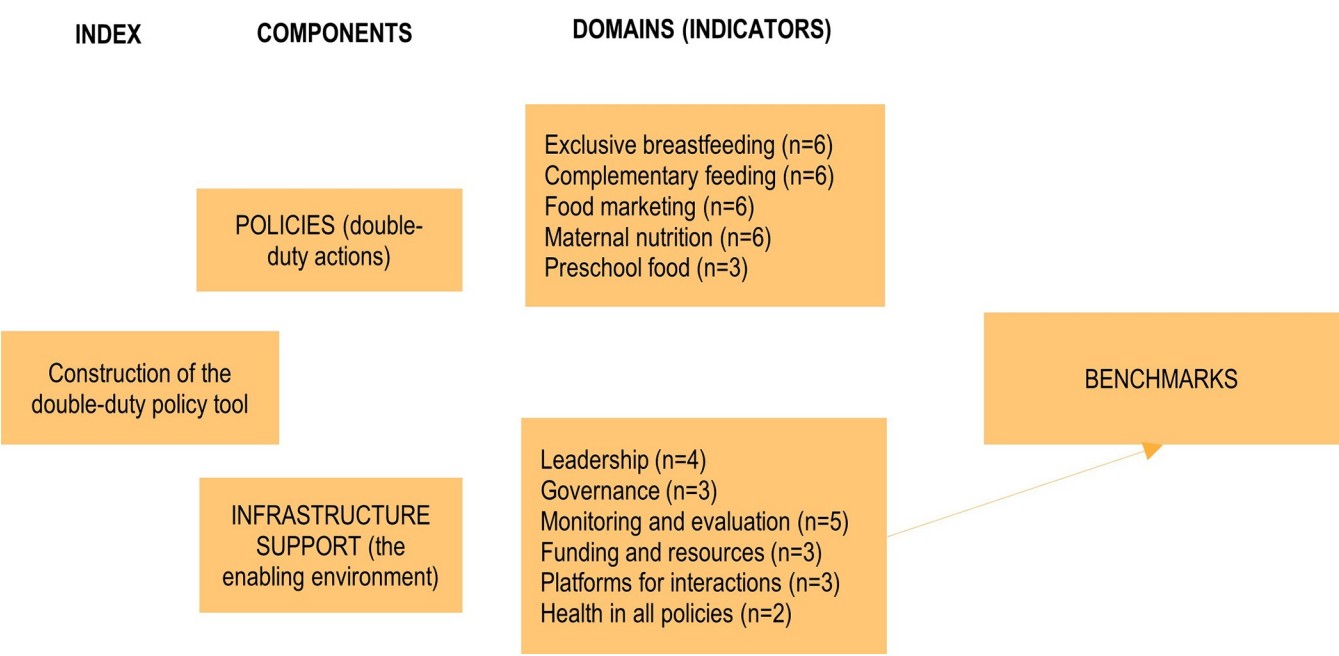

INDEX          COMPONENTS          DOMAINS (INDICATORS)

**Fig 2. Components and indicators of the double-duty policy tool used.**

period but still in effect, was considered eligible. The search period was February to April 2020. Evidence was extracted using an Excel spreadsheet including name, date of publication, enacting institution, type of evidence (legislation, policy/strategy document, policy, budget document, practical/programmatic guidance, website, strategy, programme/action), and details on relevant content (**S2 Table**). Each piece of evidence found was coded according to the indicator it referred to. We also followed up with some stakeholders (n = 4) to discuss the emerging evidence to validate the evidence gathered and/or collect further evidence. Requests for information were also submitted to relevant government authorities.

### • Step 1.3: Validating the evidence with experts

The emerging evidence was collated into a draft 'evidence paper' which was shared with relevant experts. For the validation process, the experts were asked whether there were any inaccuracies or any governmental action(s) (policy, plan, strategy, programme, technical document) omitted from the evidence document. Ten of the 16 experts suggested changes to 22 policy indicators and seven infrastructure support indicators during the validation process. These changes included: identifying additional evidence, adding further detail about some of the evidence provided, as well as removing some evidence that was considered irrelevant. The final evidence paper included evidence of action by the Government of Peru to prevent multiple forms of malnutrition in IYC for each indicator of good practice.

**Step 2. Assess the level of implementation of government actions.** The 16 experts were consulted on the degree of implementation of each of the indicators of the DDAs and infrastructure support for addressing multiple forms of malnutrition in IYC. In this step, the experts were asked to rate the degree of implementation according to the four phases of the policy implementation cycle (agenda setting, formulation, implementation, evaluation) [15].

**Step 3. Identify and prioritise government actions.** The 16 experts were asked to prioritise each of the 47 good practice indicators on a scale of 1 to 5 (where 1 = very low, 2 = low,

**Policies indicators (double-duty actions)**

**1. Initiatives to promote and protect exclusive breastfeeding (EBF) in the first 6 month, and beyond**
**BREASTFEEDING1:** National breastfeeding policy
**BREASTFEEDING2:** The international Code of Marketing of Breast Milk Substitutes
**BREASTFEEDING3:** National Baby Friendly Hospital Initiative/health facility initiative
**BREASTFEEDING4:** Paid maternity leave legislation
**BREASTFEEDING5:** National standard and guidelines for BF promotion
**BREASTFEEDING6:** Community-based breastfeeding outreach and support activities have national coverage
**2. Promotion of appropriate early and complementary feeding in infants**
**COMPFEEDING1:** National infant and young child feeding policy
**COMPFEEDING2:** Counseling for complementary feeding
**COMPFEEDING3:** Vitamin A supplementation
**COMPFEEDING4:** Iron supplementation
**COMPFEEDING5:** Multi Micronutrient (MMN) powders
**COMPFEEDING6:** Preventive zinc supplementation
**3. Regulations on marketing' double duty actions**
**MARKETINGFOOD1:** Restriction of promotion of unhealthy (TV, radio)
**MARKETINGFOOD2:** Restriction of promotion of unhealthy foods on the internet, networks, etc.
**MARKETINGFOOD 3:** Restriction of promotion of unhealthy foods in settings where children gather

**MARKETINGFORMULA1:** Restriction of promotion of infant formula (TV, radio)
**MARKETINGFORMULA2:** Restriction of promotion of infant formula on the internet, networks, etc.
**MARKETINGFORMULA3:** Restriction of infant formula promotion in ante-natl and post-natal settings
**4. Maternal nutrition and antenatal care programmes**
**MCH1:** Iron and folic acid supplementation in pregnant women
**MCH2:** Nutritional counseling for pregnant women
**MCH3:** Calcium supplementation for pregnant women
**MCH4:** Vit A supplementation for pregnant women
**MCH5:** Deworming in pregnant women residing in areas with 20% or higher prevalence of infection with hookworm or T.trichiura infection AND a 40% or higher prevalence of anemia
**MCH6:** Delayed cord clamping
**5. Pre-School food programmes and policies**
**PRESCHOOL1:** Food composition targets/standards have been established for pre-school food service outlets
**PRESCHOOL2:** Clear, consistent policies (including nutrition standards) implemented in early childhood education services for food service activities (canteens, food at events, fundraising, promotions, vending machines etc.)
**PRESCHOOL3:** Good support and training systems to help pre-school settings and their caterers meet the healthy food service policies and guidelines

**Infrastructure support indicators (the enabling environment)**

**6. Leadership**
**LEAD1:** There is strong, visible, political support (at the Head of Government / Cabinet level) for preventing obesity, stunting and iron deficiency anaemia in young children (adapted from FOOD-EPI)
**LEAD2:** Clear population intake targets have been established by the government for the nutrients of concern (iron, added sugar, saturated fat and protein to meet WHO and national recommended dietary intake levels (adapted from FOOD-EPI)
**LEAD3:** Clear, interpretive, evidence-informed food-based dietary guidelines have been established and implemented to promote healthy, diverse diets for young children (adapted from FOOD-EPI)
**LEAD4:** Government priorities have been established to reduce inequalities or protect vulnerable populations in relation to obesity, stunting and iron deficiency anemia in young children (adapted from FOOD-EPI)
**7. Governance**
**GOVER1:** There are robust procedures to restrict commercial influences on the development of policies related to preventing obesity, stunting and iron deficiency anaemia in young children (adapted from FOOD-EPI)
**GOVER2:** Policies and procedures are implemented for using evidence in the development of nutrition policies related to obesity, stunting and iron deficiency anaemia in young children (adapted from FOOD-EPI)
**GOVER3:** The government ensures access to comprehensive nutrition information and key documents (e.g. budget documents, annual performance reviews and health indicators) for the public (adapted from FOOD-EPI)
**8. Monitoring and intelligence:**
**MONIT1:** Monitoring systems, implemented by the government, are in place to regularly monitor food environments (for the five proposed double duty actions above), against codes/guidelines/standards/targets (adapted from FOOD-EPI)
**MONIT2:** There is regular monitoring of young children's nutritional status based on population level intakes against specified intake targets or recommended daily intake levels (adapted from FOOD-EPI)
**MONIT3:** There is regular monitoring of the status and progress of obesity, stunting and iron deficiency anaemia in young children using anthropometric and biological measurements (adapted from FOOD-EPI)
**MONIT4:** There is sufficient evaluation of major programs and policies to assess effectiveness and contribution to reducing obesity, stunting and iron deficiency anaemia in young children (adapted from FOOD-EPI)
**MONIT5:** Progress towards reducing nutritional inequalities in vulnerable young children's populations are regularly monitored (adapted from FOOD-EPI)

**9. Funding and resources:**
**FUND1:** Funding for interventions and policies to reduce obesity, stunting and iron deficiency anaemia in young children as a proportion of total health spending is sufficient to reduce their prevalence and reduce associated inequalities (adapted from FOOD-EPI)
**FUND2:** Government funded research is targeted for improving obesity, stunting and iron deficiency anaemia in young children and their related inequalities (adapted from FOOD-EPI)
**FUND3:** There is a statutory health promotion agency in place that includes an objective to promote healthy, diverse diets of young children, with a secure funding stream (adapted from FOOD-EPI)
**10. Platforms for interactions**
**PLATF1:** There are robust coordination mechanisms across departments and levels of government (national and local) to ensure policy coherence, alignment, and integration of policies in obesity, stunting and iron deficiency anaemia in young children policies across governments (adapted from FOOD-EPI)
**PLATF2:** There are formal platforms between government and the commercial food sector to implement healthy food policies to prevent obesity, stunting and iron deficiency anaemia in young children (adapted from FOOD-EPI)
**PLATF3:** There are formal platforms for regular interactions between government and civil society on nutrition policies and other strategies to prevent obesity, stunting and iron deficiency anaemia in young children (adapted from FOOD-EPI)
**11. Health in all policies**
**HIAP1:** There are processes in place to ensure that impacts on obesity, stunting and iron deficiency anaemia in young children in vulnerable populations are considered and prioritised in the development of all government policies relating to food (adapted from FOOD-EPI)
**HIAP2:** There are processes (e.g. health impact assessments) to assess and consider nutrition and health impacts during the development of other non-food policies (adapted from FOOD-EPI)

**Fig 3. Policy and infrastructure-support indicators (short description of indicators).**

3 = medium, 4 = high and 5 = very high). For this prioritisation exercise, two criteria were used: impact (i.e. the extent of the expected impact of the action, including the likely effectiveness in reducing the burden of malnutrition in IYC and other benefits) and feasibility (i.e. the ease with which the action can be carried out, accounting for political, budgetary and social realities).

For both steps 2 and 3, experts were asked to provide justifications for their ratings through open-ended questions.

## Study participants

Sixteen food/nutrition experts participated. Purposive qualitative sampling was used, which is appropriate when a sample of participants who best represent or know the research topic is required (16). The criteria for selecting the experts were as follows: having >10 years' experience in food/nutrition policy for IYC and be in senior management positions, with leadership and decision-making skills (20). Experts were selected for the three categories of organisations involved in the implementation of food/nutrition policy for IYC: government institutions (n = 11), non-governmental organisations (n = 3) and international cooperation agencies (n = 2) (**S3 Table**). Each expert validated the evidence document produced, rated the degree of implementation of the policies and prioritised the current and implemented policies in Peru with the potential to become DDAs addressing the coexistence of multiple forms of malnutrition in IYC. The expert consultations were conducted through individual online semi-structured interviews rather than through a deliberative panel, as is usual in Food-EPI methodology. This change was due to the social distancing imposed by the COVID-19 pandemic. A pilot test was carried out with two experts to test the questionnaires, which allowed adjustments to the wording for better understanding.

## Data management and analysis

For the quantitative analysis, frequencies and percentages were generated for the 47 indicators according to the rating on the level of implementation given by experts (agenda setting/formulation/implementation/evaluation). A graph was produced showing the proportion of experts who classified government actions in agenda setting/formulation vs. implementation/evaluation phases. The cut-off point for deciding whether actions were in early or later stages of implementation was $\geq$ 50%. For the prioritisation of DDAs and infrastructure support, mean scores were created for impact and feasibility. Cut-off points based on mean scores were defined for high impact ($\geq 4$) and high feasibility ($\geq 3$). A graph was created using the mean scores and the cut-off points classifying the indicators into four groups (high impact and high feasibility; low impact and low feasibility; low impact and high feasibility; high impact and low feasibility).

For the qualitative verbatim data, deductive content analysis was conducted in three main phases (*preparation*, *organisation*, and *reporting* [16,17]). This approach is used when the structure of the analysis is based on prior knowledge. Content analysis was conducted with a focus on trustworthiness at each phase [19]. In the *preparation* phase, two issues of interest guided the codebook development (**S4 Table**): 1. Barriers to implementing DDAs and infrastructure support indicators; and 2. Reasons for prioritisation of DDAs and infrastructure support indicators. In the *organisation* phase, data reduction was conducted by organising and categorising the data according to the pre-defined codebook. Two guiding questions were considered in the *reporting* phase: RQ1: What are the barriers to low implementation (agenda setting/formulation stage) of the DDAs and infrastructure support indicators; RQ2: What were the reasons given by stakeholders to justify the prioritised DDAs

and infrastructure support indicators (high impact/feasibility). The analysis was conducted in NVivo version 12.

**Ethical approval.**   Ethical approval for the PERUSANO project was obtained from the Ethics Review Committee of the Instituto de Investigación Nutricional (IIN) Peru (reference 388-2019/CIEI-IIN). As the study was conducted during the period of mandatory confinement due to the COVID-19 pandemic, all participants gave verbal informed consent after receiving written (by email) and verbal (by virtual interview) information about the study. Verbal consent was obtained from the participant via virtual interview and this process was recorded and archived in the study's regulatory folder. The procedures for taking verbal informed consent were included in the research protocol and were evaluated and approved by the Ethics Review Committee of the Instituto de Investigación Nutricional (IIN) Peru (reference 388-2019/CIEI-IIN).

## Results

### Step 1- Mapping of current policy actions: Double-duty actions and 'infrastructure support'

**Size and extent of the evidence.**   The summary evidence document prepared in step 1 of this study synthesises all the evidence found for the implementation of government actions for each of the 47 indicators set out in the constructed tool [18]. Overall, 143 policy documents were included and the number of documents for each indicator varied (**S1** and **S2 Figs)**. It should be noted that the evidence collected refers to government actions designed to address stunting, anaemia and overweight in IYC as specific problems, since there is still no explicit policy for addressing the DBM in Peru.

**Evidence of current government actions: 'Double-duty actions'.**   The DDA indicators with the most evidence (7–10 records each) were: *complementary feeding counselling, iron supplementation for children, multi-micronutrient powders (MMN) for children and paid maternity leave legislation*. The DDA indicators with the least (1 record) evidence were: *deworming of pregnant women, International Code of Marketing for breastmilk substitute and national policy on infant feeding*. For the action "*Food composition targets/standards have been established for pre-school food service outlets,*" evidence was found only for the Cuna Más programme (government programme that aims to improve the development of children under 36 months of age in areas of poverty and extreme poverty). For the action *"The government ensures that there are clear, consistent policies (including nutrition standards) implemented in early childhood education services for food service activities (canteens, food at events, fundraising, promotions, vending machines etc.) to provide and promote healthy food choices"*, the evidence identified was focused on the promotion rather than provision of healthy food options (see S1 Fig).

**Evidence of current government actions: 'Infrastructure support'.**   Within the infrastructure support domains, most evidence (range 8–12 records) was identified for: *access to comprehensive nutrition information and key documents; robust coordination mechanisms across departments and levels of government to ensure policy coherence, alignment and integration of policies* (considering the formal coordination structures created to address stunting and anaemia in IYC); and *political (executive) support*. The action with the least evidence (1 record) was *restricting commercial influences on the development of policies related to DBM in IYC*. For *formal platforms between government and the commercial food sector to implement healthy food policies to prevent DBM in IYC*, no specific records were found (S2 Fig).

## Step 2. Assess the level of implementation of government actions

**Extent of implementation of government actions: 'Double-duty actions'.** As shown in **Fig 4**, the Peruvian government has made some effort to implement policies that have the potential to become DDAs, but only 5/27 DDAs indicators were seen as fully implemented by all national experts (*International Code of Marketing for breastmilk substitutes, paid maternity*

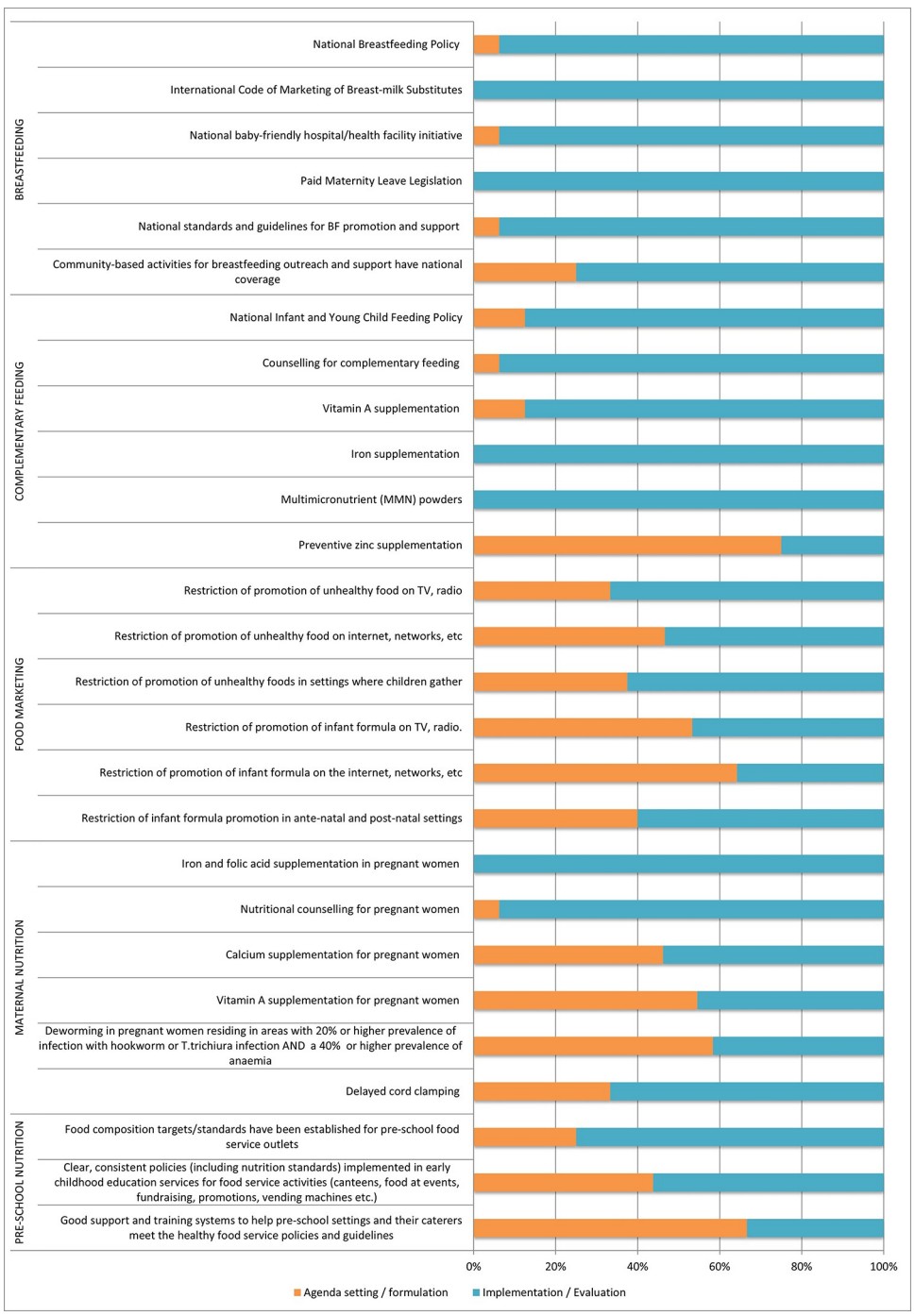

**Fig 4. Extent of implementation of double-duty actions in relation to the policy cycle.**

*leave legislation, iron supplementation for children, MMN for children, nutritional counselling for pregnant women).* Slightly more than half (16/27) of the DDAs indicators were rated at the full implementation/evaluation stage by ≥ 50% (n≥ 8) of the experts. However, 6/27 DDA indicators were classified in the early stages of the policy cycle (agenda setting/formulation) by ≥50% (n≥ 8) of experts: *preventive zinc supplementation for children (75%); good support and training systems to help pre-school settings and their caterers meet the healthy food service policies and guidelines (67%); restricting promotion of infant formula on the internet, networks, etc (64%); deworming in pregnant women residing in areas with 20% or higher prevalence of infection with hookworm or T. trichiura infection and a 40% or higher prevalence of anaemia (58%); vitamin A supplementation for pregnant women (55%); and restriction of promotion of infant formula on TV, radio (53%).*

**Identified barriers for government actions: 'Double-duty actions' with low implementation.** For *preventive zinc supplementation*, experts identified the lack of specific regulations or policy as a barrier:

"*If zinc appears in any standard, it is because it is included in the multi-micronutrient package, but that standard is aimed at reducing anaemia. It is not because there is a definite policy of preventive zinc supplementation.*" (National representative, non-governmental organisation).

The experts pointed out that the implementation of *preventive zinc supplementation* in children was evaluated but there was insufficient evidence to consider zinc deficiency in children as a public health problem to justify implementation.

"*We should be careful because when preventive supplementation was technically discussed, the analysis was that there was not enough evidence in the country for us to implement a preventive zinc supplementation scheme. The fact that we do not have a preventive zinc supplementation strategy for children is not wrong, it is just that it is not needed.*" (National representative, international agency)

Experts also mentioned the gap in monitoring and evaluation as a major barrier to *good support and training systems to help pre-school settings.* Experts again reported that regulations exist but were unaware of any monitoring of their implementation or evaluation.

"*What exists is the regulation, but there is no evidence of evaluation or monitoring or whether it is properly executed; as a general structure there is none.*" (National representative, international agency)

Weak implementation, monitoring, and evaluation processes; legal loopholes and sanction processes were identified as barriers for *restricting the promotion of infant formula on TV and radio.* Participants mentioned that even though regulations exist, there were difficulties in implementing them. Also, experts reported that weak enforcement and legal loopholes prevented companies from sanction for non-compliance.

"*The regulation is there, but between the lines you can find ways to evade this, and that in fact happens because there are still advertisements on TV.*" (National representative, government sector)

Limited operational capacity to implement sanctions and weak monitoring/evaluation were identified as obstacles for *restricting the promotion of infant formula on the internet and social*

*networks*. Experts commented that according to current regulations, such a policy should be implemented, but there is no evidence of its implementation. Other experts mentioned how difficult it is to regulate what happens on the internet and social networks.

"*This is more difficult to control, we are far beyond what really happens on the internet.*" (National representative, international agency)

For *deworming in pregnant women*, participants also reported gaps in implementation, monitoring, and evaluation as a barrier.

"*The norm exists, the implementation happens very much by the hand of someone who remembers and does it. There is a weak implementation part and there is no evaluation.*" (National representative, government sector)

For *vitamin A supplementation for pregnant women*, experts did not identify specific barriers to its implementation, as they considered it unnecessary. Experts indicated that Vitamin A supplementation was given to postpartum women until 2014, when it was discontinued. According to the experts interviewed, Vitamin A supplementation in pregnant women is unnecessary in Peru and they also argued that the evidence suggests that uncontrolled Vitamin A supplementation during pregnancy could be teratogenic, which is why it is not on the agenda of the Ministry of Health.

"*There is no vitamin A supplementation in pregnant women, because by definition the evidence suggests that excess vitamin A can be teratogenic, that is why it is not given to pregnant women.*" (National representative, non-governmental organization)

**Extent of implementation of government actions: 'Infrastructure support'.** In relation to the infrastructure support indicators (Fig 5). Only 1/20 infrastructure support indicator (*Government ensures access to comprehensive nutrition information and key documents*) was rated as fully implemented by all 16 experts. Almost three quarters (14/20) of infrastructure support indicators were rated as fully implemented by ≥50% (n = 8) of experts. Six (of 20) infrastructure-related actions were ranked in the early stages (agenda setting/formulation) by ≥50% (n = 8): *Formal platforms between government and the commercial food sector to implement healthy food policies to prevent DBM in IYC (100%); restrict commercial influences on the development of policies related to DBM in IYC (85%); statutory health promotion agency in place that includes an objective to promote healthy, diverse diets of young children, with a secure funding stream (71%); processes to assess and consider nutrition and health impacts during the development of other non-food policies (69%); establishment of nutrient intake targets (iron, added sugar, fats) (57%); government funded research to target DBM in IYC and related inequalities (50%).*

**Identified barriers for government actions: 'Infrastructure support' with low implementation.** The barriers to *introducing a formal platform between government and the commercial food sector to implement healthy food policies to prevent DBM in IYC* was seen to be due to a lack of regulation or existing policy and political constraints due the lack of continuity of government initiatives during changes of government in the country, as an expert explained:

"*A working group was implemented at some point with the food sector CONFIEP [National Confederation of Private Business Institutions], but when there were changes of government,*

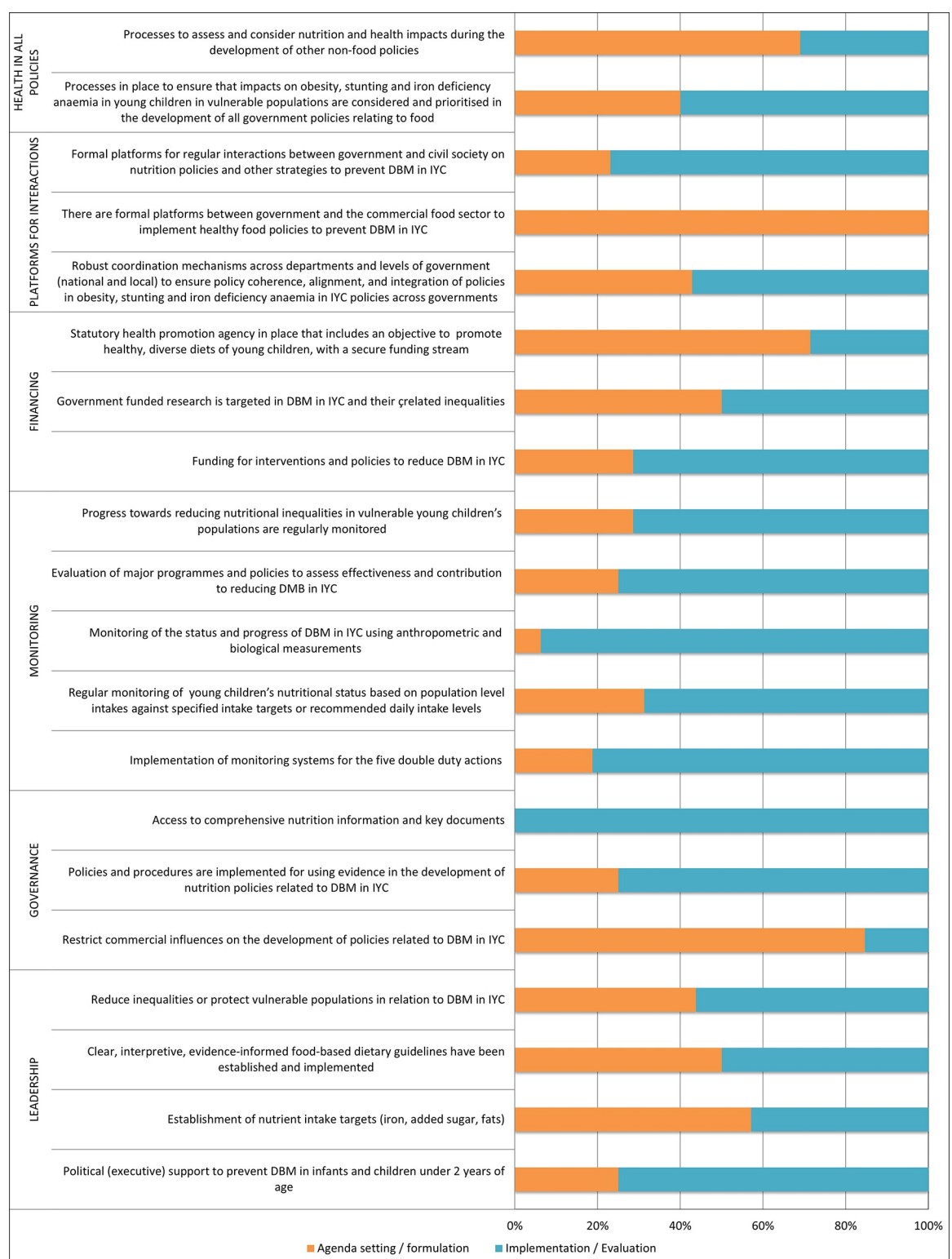

**Fig 5. Extent of implementation of infrastructure support domains, in relation to the policy cycle.**

*it was deactivated and there was no longer that joint work with the food sector like the one that had started in 2018, political changes affect the course of public policies, unfortunately.*" (National representative, government sector)

Barriers encountered for *restricting commercial influences on the development of policies related to preventing obesity, stunting and iron deficiency anaemia in young children* were lack of regulation or policy for a structured procedure, and the existence of commercial influences on policy making, as these experts mentioned:

"*The national industry society has lobbied congress not to enact the healthy eating act, but the government has not implemented strong procedures to curtail this lobbying. There are no laws to combat the double agenda, or the revolving door of congresspeople with officials of food manufacturing companies.*" (National representative, non-governmental organisation)

"*The government is doing nothing to prevent the private sector from overtaking it, there is nothing on the public agenda. You must look at examples from other countries. The legislative apparatuses of the countries have legal devices that require candidates to declare who finances their candidacy and to make transparent the so-called management of interests, which is fully regulated and not hidden. Research on the formalisation and transparency of political work is a segment of the tasks that does not exist in Peru.*" (National representative, international organisation)

Barriers to implementing a *statutory health promotion agency in place that includes an objective to promote healthy, diverse diets of IYC with a secure funding stream* were insufficient budgets, due to the political instability of the country and the lack of sustainable funding.

"*The budget is always too small for health promotion actions. This is even more accentuated when there is instability.*" (National representative, government sector)

Experts identified barriers *to processes to assess and consider nutrition and health impacts during the development of non-food policies*, such as a lack of a health impact assessment, as this process was inadequately monitored and supervised.

For *establishment of nutrient intake targets (iron, added sugar, fats)*, the experts identified lack of regulation or policy as a barrier, and indicated that it is not currently on the government's agenda to implement them, as experts explained:

"*Ideally, we would like to implement it, but unfortunately I feel that the government has not set clear targets, they have only set the parameters; only in the case of iron this is in the formulation phase.*" (National representative, international organisation)

"*This is extraordinarily complex to implement because we know how much iron a child needs, but it is not covered, that is why we must cover it, because we know that the diet is deficient. What we have done here is to implement front-end labelling, which is a different way of having clear targets for reducing intake. So far, we have not worked with intake targets in the country. However, it is not something that can be done now.*" (National representative, government sector)

Barriers for *government funded research targeting obesity, stunting and iron deficiency anaemia in IYC and their related inequalities*, included insufficient government-supported research and the budget gap between different institutions, as one expert said:

"*In general, research in the country has been improving a lot in terms of the amount of money that is invested, but it is still small compared to other countries like Colombia, Mexico, Canada, Chile and Brazil.*" (National representative, international agency)

### Step 3. Identify and prioritise government actions

**Priority government actions: 'Double-duty actions'.** A total of 12 indicators across the 27 DDAs were identified and prioritised as "high impact" and "high feasibility" (**Figs 6 and 7**). Five actions focused on breastfeeding (*breastfeeding policy, baby-friendly hospital/health facility, paid maternity leave, standards/guidelines from breastfeeding promotion/support, community breastfeeding support*). Three actions were around maternal nutrition (*iron/folic acid supplementation, nutritional counselling, delayed cord clamping*). Two actions were linked to complementary feeding (*counselling for complementary feeding and iron supplementation*). Lastly, one action was prioritised from food marketing (*restrict promotion of unhealthy foods in children's settings*) and from preschool nutrition (*consistent policies/nutrition standards*).

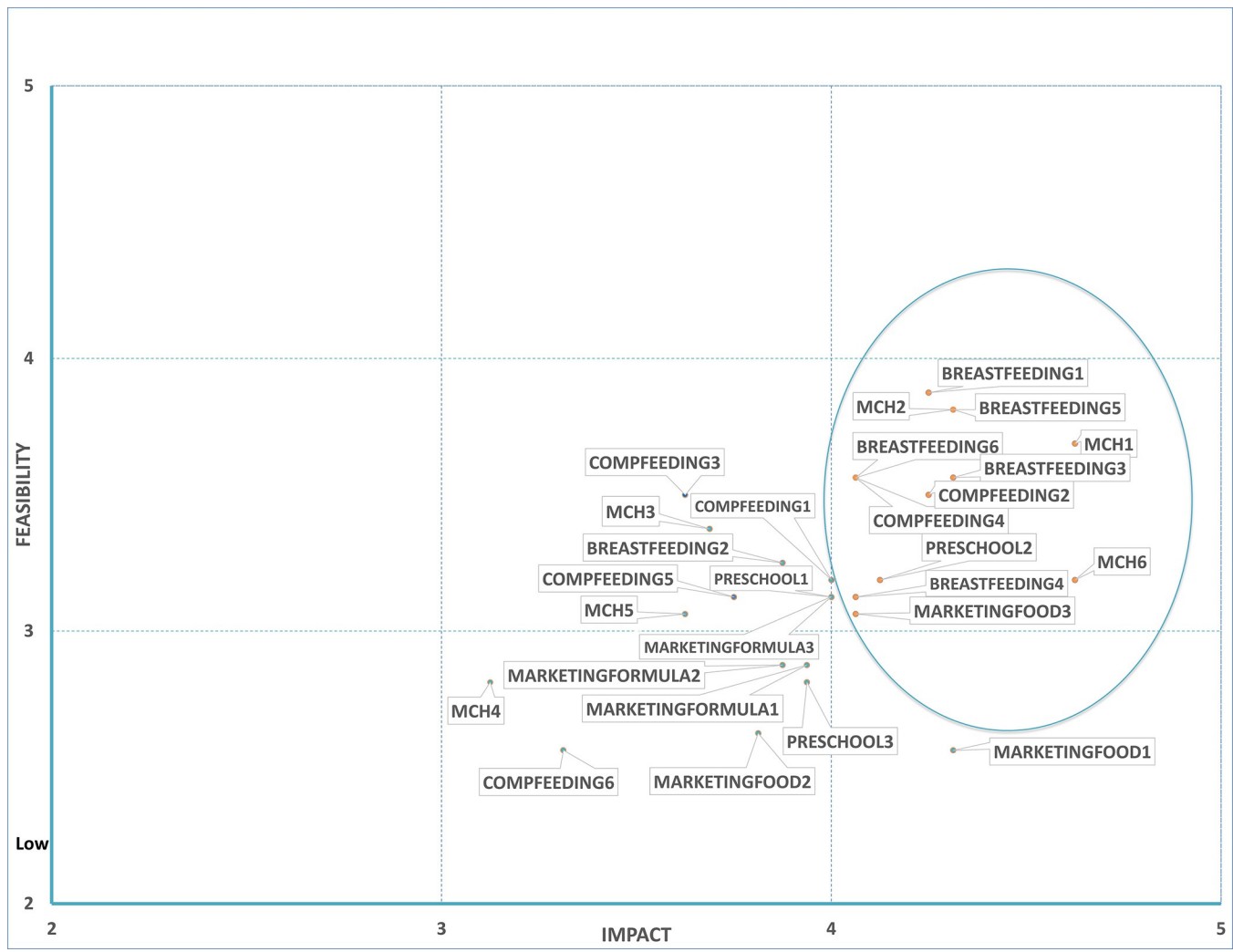

**Fig 6. Expert ratings of expected impact against feasibility of double-duty actions.**

| Double-duty actions prioritised as of higher impact and feasibility | | |
|---|---|---|
| | Indicator code | short description of indicators |
| | BREASTFEEDING1 | National Breastfeeding Policy |
| | BREASTFEEDING3 | National baby-friendly hospital/health facility initiative |
| | BREASTFEEDING4 | Paid maternity leave legislation |
| | BREASTFEEDING5 | National standards and guidelines for BF promotion and support |
| | BREASTFEEDING6 | Community-based activities for breastfeeding outreach and support have national coverage |
| | COMPFEEDING2 | Counselling for complementary feeding |
| | COMPFEEDING4 | Iron supplementation |
| | MARKETINGFOOD3 | Restriction of promotion of unhealthy foods in settings where children gather |
| | MCH1 | Iron and folic acid supplementation in pregnant women |
| | MCH2 | Nutritional counselling for pregnant women |
| | MCH6 | Delayed cord clamping |
| | PRESCHOOL2 | The government ensures that there are clear, consistent policies (including nutrition standards) implemented in early childhood education services for food service activities to provide and promote healthy food choices (note: this the only action in early implementation stage) |

**Fig 7. Short description of double-duty actions prioritised.**

**Priority government actions: 'Infrastructure support'.** A total of eight indicators across the 20 infrastructure support indicators were identified and prioritised as "high impact" and "high feasibility" to prevent DBM in IYC (**Figs 8 and 9**). This included three actions for leadership (*political support to prevent DBM in IYC; food-based dietary guidelines, protect vulnerable IYC*); three actions for monitoring (*implement monitoring for all DDAs; monitor the progress of DBM in IYC; monitor nutritional inequalities in vulnerable IYC*); and one action from each of governance (*procedures implemented for evidence use in policy development*) and funding/resources (*funding for interventions*).

**Reasons for prioritisation of government actions: 'Double-duty actions' and 'infrastructure support'.** Established regulation, easy implementation, available budget, and relevance

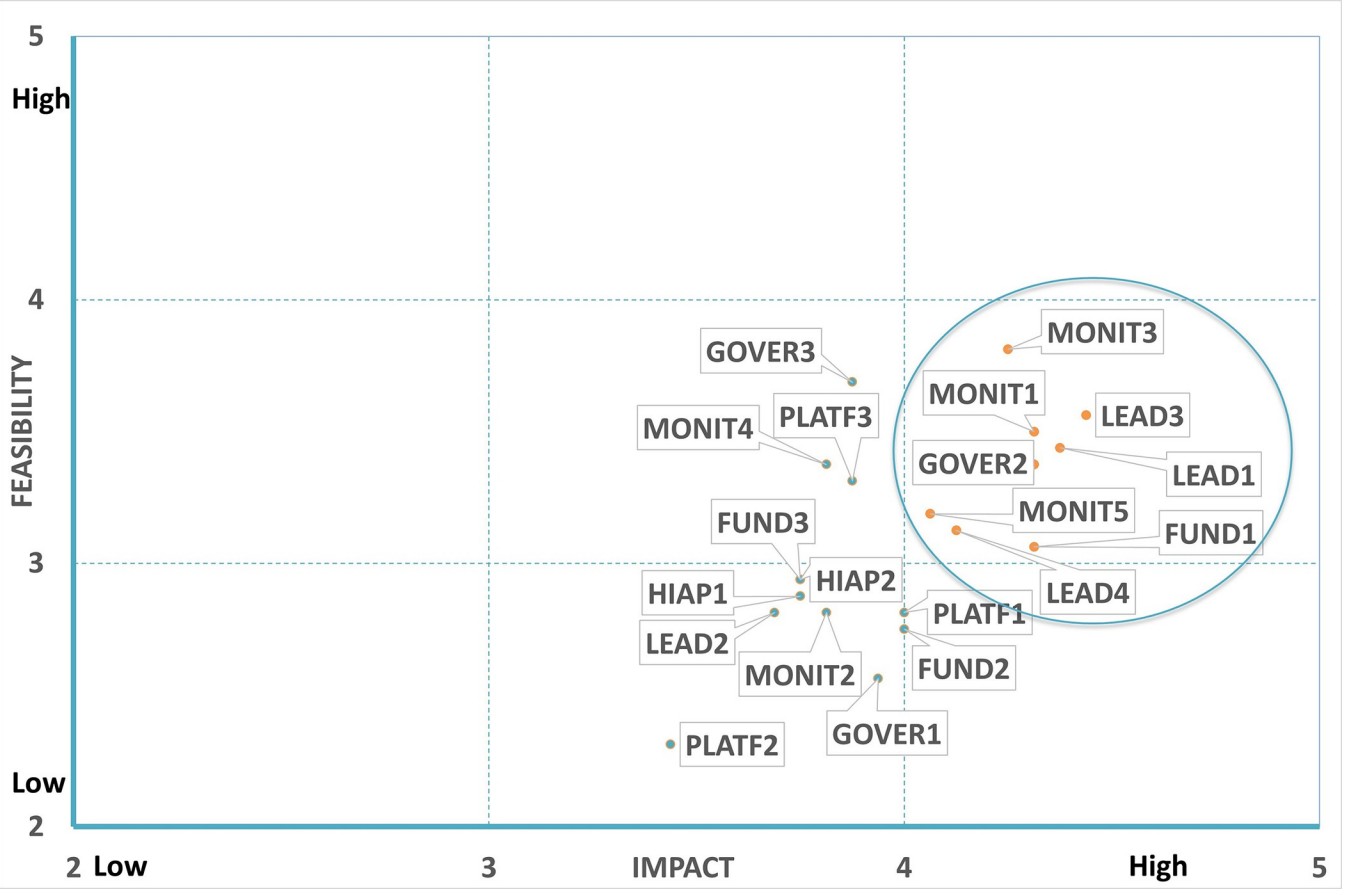

**Fig 8. Expert ratings of expected impact against feasibility of infrastructure support indicators.**

were the most frequent reasons given by experts when prioritising double-duty actions and infrastructure support indicators (**S5 and S6 Tables**).

## Discussion

### Summary of findings

This study assessed current implementation of- and priority for- government-level actions to tackle multiple forms of malnutrition in IYC in Peru against 47 indicators of good practice for five DDAs and for the enabling policy environment, i.e. 'infrastructure support'.

**Policy actions that are well implemented.**   Several policy actions were well implemented such as: iron and folic acid supplementation for pregnant women; iron supplementation for children; multi-micronutrient powders; paid maternity leave legislation; and the international code of marketing for breastfeeding substitutes. Furthermore, about half of double-duty actions and almost three quarters (14/20) of the infrastructure support indicators were categorised at full "implementation/assessment" stage.

These results can be explained by the fact that, over the last decades, the Peruvian government has demonstrated a strong political commitment to successfully addressing stunting. Evidence shows that the combination of four key actions helped reduce the prevalence of stunting among children: the implementation of the National Nutrition Strategy (known as CRECER), the creation of the budget programme called the Articulated Nutrition Programme (called

| | Infrastructure support indicators prioritised as of higher impact and feasibility | |
|---|---|---|
| **Indicator code** | **short description of indicators** | |
| 📄 | LEAD1 | Political (executive) support to prevent DBM in infants and children under 2 years of age |
| 📄 | LEAD3 | Clear, interpretive, evidence-informed food-based dietary guidelines have been established and implemented (note: this the only action in early implementation stage) |
| 📋 | LEAD4 | Reduce inequalities or protect vulnerable populations in relation to obesity, stunting and iron deficiency anaemia in young children |
| 🏛 | GOVER2 | Policies and procedures are implemented for using evidence in the development of nutrition policies related to obesity, stunting and iron deficiency anaemia in young children |
| 🖥 | MONIT1 | Implementation of monitoring systems for the five double duty actions |
| 🖥 | MONIT3 | Monitoring of the status and progress of obesity, stunting and iron deficiency anaemia in young children using anthropometric and biological measurements |
| 🖥 | MONIT5 | Progress towards reducing nutritional inequalities in vulnerable young children's populations are regularly monitored |
| 💰 | FUND1 | Funding for interventions and policies to reduce obesity, stunting and iron deficiency anaemia in young children |

**Fig 9. Short description of infrastructure support indicators prioritised.**

PAN), the adoption of a conditional cash transfer programme (known as JUNTOS) and the Child Nutrition Initiative (made up of non-governmental organisations, international cooperation agencies and civil society organisations) [19]. It is evident that strong coordination mechanisms between departments and levels of government to ensure coherence, alignment and integration of policies was key. The present study shows that this infrastructure support action was one of the most evidenced government actions identified (S2 Fig) (considering the formal coordination structures created to address stunting and anaemia in IYC). All these key actions, knowledge, and expertise in reducing stunting could be harnessed to address DBM. Currently, anaemia and overweight/obesity continue to be a challenge for Peruvian public health services despite the government's efforts to reduce them.

**Policy actions that need strengthening.** Nearly a quarter (6/27) of DDA indicators were classed in the early stages of the policy cycle such as: zinc supplementation; support/training for preschools and their food providers; restricting promotion of infant formula on both the

internet/social media and on TV/radio. Experts considered that only one-third (6/20) of the infrastructure support indicators were at the full "implementation/evaluation" stage. Furthermore, there was no evidence of government action in relation to formal platforms between government and the commercial food sector to implement healthy food policies.

These findings could be explained by the fact that Peru does not yet have an explicit policy to address the DBM, and in the present study experts identified obstacles to the implementation of DDAs and infrastructure support as: lack of regulations, inadequate monitoring/evaluation to ensure compliance, commercial influences on policy makers, insufficient resources, changing public health priorities with the COVID-19 pandemic, and political instability.

## Government actions with low implementation

Among the DDA indicators (6/27) categorised in the early stages of the policy cycle, three of these were linked to undernutrition (zinc/vitamin A supplementation and deworming). Firstly, experts believed that preventive zinc supplementation for children was irrelevant as a public health policy in Peru due to the absence of current data on zinc deficiency. However, in 2011, 34.5% of the Peruvian population was estimated to be deficient in zinc intake based on dietary intake data [20]. Despite Vitamin A deficiency concerns in rural Peru [21], Vitamin A supplementation is low in pregnant women, which can be explained in part by worries raised in the qualitative analysis by experts regarding the teratogenicity of high doses. Lastly, implementing deworming policy during pregnancy was weak despite the existence of a directive [22]; barriers to implementation included poor monitoring and evaluation.

Other actions categorised in the early stages of the policy cycle related to marketing of infant formula in online media or TV/radio and training caterers in pre-school settings. Experts identified several barriers to implementing restrictions on promoting infant formula on the internet, TV, and radio: monitoring, evaluation and sanctioning processes were seen as weak and legal loopholes hindered their implementation. An analogous situation exists in other countries, for example in Mexico, mothers of IYC are highly exposed to marketing of breastmilk substitutes through TV, radio and social media, even though Mexican legislation prohibits its promotion [23]. The WHO indicates that most countries in the Americas region have legislation prohibiting at least some form of breastmilk substitute promotion [24]. However, marketing persists, as there are gaps in legislation where conflicts of interest, some promotional mechanisms and public announcements are not fully covered by existing legislation. The WHO highlights the importance of political will at the highest level to accelerate progress in the region.

We found that the implementation of support and training systems for caterers in pre-school settings was poor. Experts highlighted the lack of monitoring and evaluation as the main barrier. Similar results were reported in studies that analysed the level of healthy food environment implementation using the Food-EPI tool in several other countries [25–28].

The present study also found that 6/20 indicators relating to infrastructure support were rated in the early stages (agenda setting/formulation) by more than half of the experts. These results are similar to those reported for Chile, which reported that 21% of infrastructure support indicators were rated as "very low/no" implementation [29]. On the other hand, countries such as Guatemala and Senegal reported the highest proportion of indicators were categorised as "low implementation" 95% (23/24) and 90% (20/22), respectively [27,28].

## Prioritisation of government actions

Across the five DDAs, 12/27 good practice indicators were prioritised to address malnutrition in IYC in Peru as they were rated as "high impact" and "high feasibility" by experts. Among

these, five relating to breastfeeding were prioritised, with reasons such as: relevance for addressing DBM, established regulations, ease of implementation and available budget.

According to a recent national survey in Peru [7], breastfeeding among children <6 months has decreased by 4.4% between 2020 (68.4%) and 2021 (64.0%), suggesting that there are still gaps to be filled to ensure breastfeeding thrives. Experts highlighted evidence of a national breastfeeding policy, as the Ministry of Health's breastfeeding committee has identified the need to implement a breastfeeding promotion law to support breastfeeding because existing regulations are often weak, especially regarding breastmilk substitutes. Experts reported that implementing mother and baby friendly hospitals needed intensive and consistent work. Experts highlighted inequalities of access to paid maternity leave for women working in the informal sector, despite legislation. Experts indicated that there is a slow process of implementation of 'national standards and guidelines for breastfeeding promotion and support' from the time they are promulgated to their full implementation within healthcare settings. The experts also highlighted the likely high impact of community-based breastfeeding activities; however, they mentioned the need for training of health staff to ensure implementation. All these aspects are important to consider to improve breastfeeding rates in Peru.

The actions prioritised concur with those from a recent review on effective maternal and child health interventions to prevent the DBM, which identified the following priority actions: breastfeeding counselling for mothers across settings, maximising breastfeeding support through legislation, support for maternity leave, workplace facilities for breastfeeding and restrictions on marketing of breastmilk substitutes [30]. The review also suggests that educational interventions delivered through community health workers during the postnatal period were effective at improving breastfeeding practices. This could be considered as a DBM indicator in future appraisals.

Three actions related to maternal nutrition were prioritised (iron and folic supplementation in women, nutritional counselling for pregnant women and umbilical cord cutting). Firstly, iron supplementation for pregnant women was seen as relevant, with available funding and established regulations. It is worth noting that experts mentioned the need for better control of compliance with supplementation to during pregnancy. Secondly, nutritional counselling for pregnant women was prioritised because of its relevance and existing regulations. Experts pointed out different aspects needed to improve its impact, such as workforce capacity to carry out counselling, train health personnel, standardise key messages, follow-up, structure, and appropriate space. Thirdly, timely umbilical cord cutting was also prioritised because of its perceived relevance, cost-effectiveness, and ease of implementation, but experts noted that success depends on staff motivation and confidence.

Two actions related to complementary feeding were prioritised (counselling for complementary feeding; iron supplementation) for several reasons: relevance, cost-effectiveness, and available budget at national, regional, and local level. This prioritisation of counselling is consistent with wider evidence (e.g. Lassi et al [31]) indicating the impact of complementary feeding education for increased weight-for-age Z-score by 0.41 standard deviations (SD) and height-for-age Z-score by 0.25 SD in IYC. The experts pointed out that there are serious difficulties inherent to the health system that hinder the optimal implementation of complementary feeding counselling, which goes beyond the capabilities/attitude of health personnel, time and available human resources, the demands of meeting targets associated with service coverage, rather than oriented to impact. A similar situation was reported in Brazil in relation to their Breastfeeding and Complementary Feeding Strategy, where the lack of specific funding, monitoring of breastfeeding and complementary feeding practices were the main challenges [32].

Two further priority actions that emerged related to nutrition policy (pre-school nutrition policy; restricting unhealthy food marketing to children). Evidence from other studies suggests that in order to enhance the cost-effectiveness of pre-school nutrition policy, it is necessary to prioritise implementation in poorer areas, closely supervise and control the supply and distribution chain, strengthen families' capacities in IYC feeding, offer palatable and culturally acceptable foods, and provide a moderate to high percentage of the recommended dietary intake of energy and key micronutrients [33]. The reasons given for prioritising unhealthy food marketing to children was because there are regulations in force and that implementation was relatively feasible. These findings are similar to those from Senegal [28] and six other countries in the Global South (Chile, Guatemala, Malaysia, Mexico, South Africa, Thailand), where the restriction of the promotion of unhealthy foods to children (via broadcast media, non-broadcast media and places where children gather) was among the political actions prioritised [29].

Eight infrastructure support indicators were prioritised. Three of these related to strengthening leadership (political support, food based dietary guidelines, reduce nutritional inequalities); three focussed on introducing better monitoring systems (monitor food environments against guidelines/standards; monitor children's weight, height and iron deficiency anaemia; monitor progress towards reducing nutritional inequalities in vulnerable IYC). The remaining indicators aimed for better governance (in developing nutrition policies for IYC) and providing more funding for interventions and policies to reduce multiple forms of malnutrition in young children, with a focus on reducing inequalities. Similar infrastructure actions were prioritised by experts in six other Global South countries (Chile, Guatemala, Malaysia, Mexico, South Africa, Thailand) to promote a healthy food environment [29], among them were: increased political support, monitoring of the nutritional status of the population, use of evidence in the development/implementation of policies, and an increase in funds for the promotion of the population's nutrition. In Ghana [25] and Kenya [34], in-country experts identified and prioritised actions to promote a healthy food environment using the Food-Epi tool. Both countries prioritised similar actions to those in Peru to address the DBM, including the need to monitor the healthiness of the food environment.

Overall, the experts in Peru highlighted difficulties that need to be taken into consideration to facilitate the implementation of the prioritised actions. For example, lack of legal feasibility/regulations, inadequate monitoring/evaluation/enforcement, commercial lobbying of policymakers, insufficient resources, competing public health priorities and political instability. In relation to the lack of resources, experts highlighted the serious shortage of health personnel, especially in nutritional counselling. To this end, the use of digital educational resources is suggested to reduce the burden on staff and educate caregivers and families. Studies show that e-& mHealth interventions could be effective in promoting healthy diets in low- and middle-income countries (LMIC) [35].

## Strengths and limitations

The first strength of this study is the production of a recent and comprehensive evidence document as a result of the policy mapping carried out. The evidence document compiles 143 regulations showing governmental actions related to DDAs and infrastructure support indicators. Another strength of this study is the incorporation of indicators aimed at tackling multiple forms of malnutrition in the policy component of the adapted Food-EPI tool. This is an important contribution since there are no existing tools that focus on multiple forms of malnutrition. This tool can be used by researchers and policymakers from other countries to appraise policies aimed at addressing the DBM in IYC. We suggest reviewing and adjusting the indicators

(include/remove/modify) to the situation of each country. For example: micronutrient deficiency can be of a different nature in each country. It is also suggested to consider the inclusion of other relevant indicators (after analysis of the available scientific evidence and the specific context of each country), such as maternal nutrition during the breastfeeding period.

Another strength of the study was the inclusion of a qualitative component to complement the quantitative rating of level of implementation and prioritisation, by providing an in-depth analysis of the barriers that hinder the implementation of government actions and of the reasons behind the prioritisation.

This study also has some limitations, firstly it was necessary to adapt the Food-EPI tool, which was developed primarily for the prevention of DR-NCDs, this fact limited the comparability of our results outside of Peru. The second limitation relates to replacing face-to-face deliberative panels usually used in the Food-EPI methodology with individual online interviews due to the COVID-19 pandemic. Whilst conducting face-to-face deliberative panels with experts may have contributed positively to the consensus process by bringing a group of experts together to prioritise government actions in the fight against malnutrition, the individual interviews allowed a deeper exploration of experts' views.

## Conclusion

We found that the Peruvian government is implementing policies related to the five double-duty actions recommended by the WHO to address the DBM from the first years of life. It should be noted that the government actions implemented were designed to address specific types of malnutrition, such as chronic childhood undernutrition, anaemia, overweight and obesity. Aligning existing actions within a national policy with specific objectives and goals aimed at addressing the DBM is recommended.

Some DDAs and infrastructure support actions need to be reinforced as they are in the early stages of the policy implementation cycle. Identified barriers to their implementation need to be addressed to improve the implementation, and therefore the impact, of these actions on the DBM. These include legal feasibility or lack of regulations, inadequate monitoring/evaluation to ensure enforcement, commercial influences on policymakers, insufficient resources, shifting public health priorities with the COVID-19 pandemic and political instability.

Twenty priority government actions were identified to address the DBM in IYC in Peru. A positive aspect is that only one of these (food service policies/ nutrition standards in early childhood education services) was assessed in the early stages of the policy implementation cycle. Another positive point is that the experts identified aspects that could strengthen the implementation of these prioritised actions, such as established regulations, allocated budget, and easy implementation.

Finally, the tool developed could be of use for other researchers interested in policy analysis for addressing the DBM in IYC. For policymakers, non-governmental organisations and civil society, the study's findings could facilitate the process of political advocacy for the DBM in IYC on the public agenda and support decision-making on strengthening and/or implementing government priority actions, through a participatory process, to prevent DBM from the first years of life.

## Supporting information

**S1 Table. Policy and infrastructure-support indicators (full description of indicators).**
(PDF)

**S2 Table. Excel spreadsheet used to extract evidence.**
(XLSX)

**S3 Table. Participants of stakeholder groups.**
(TIFF)

**S4 Table. Codebook for analysis.**
(DOCX)

**S5 Table. Reasons for prioritisation of double-duty actions.**
(DOCX)

**S6 Table. Reasons for prioritisation of infrastructure support domains.**
(DOCX)

**S1 Fig. Summary of evidence: Number of documents against each indicator of good practice for the five double-duty actions.**
(TIFF)

**S2 Fig. Summary of evidence: Number of documents against each indicator of good practice for the infrastructure support domains.**
(TIFF)

## Acknowledgments

We thank all the participants who took part in the online interviews for their valuable contributions, as well as the support in transcribing the interviews and editing graphics and tables by six undergraduate nutritionists (Yuliza Lara Romero, Mirella Carrillo and Deysi Lozano, Adela Alcedo, Marcos Chávez and Mariano Gallo) and one undergraduate in childhood education (Jasmin Vera) from UNMSM.

## Author Contributions

**Conceptualization:** Rebecca Pradeilles, Hilary M. Creed-Kanashiro, Emily Rousham, Edwige Landais, Emma Haycraft, Michelle Holdsworth.

**Data curation:** Violeta Magdalena Rojas Huayta.

**Formal analysis:** Violeta Magdalena Rojas Huayta, Rebecca Pradeilles.

**Funding acquisition:** Emily Rousham, Rossina Pareja.

**Investigation:** Violeta Magdalena Rojas Huayta, Rebecca Pradeilles, Hilary M. Creed-Kanashiro, Emily Rousham, Doris Delgado, Edwige Landais, Nervo Verdezoto, Emma Haycraft, Michelle Holdsworth.

**Methodology:** Rebecca Pradeilles, Michelle Holdsworth.

**Project administration:** Rossina Pareja.

**Supervision:** Rebecca Pradeilles, Hilary M. Creed-Kanashiro, Emily Rousham, Rossina Pareja, Michelle Holdsworth.

**Writing – original draft:** Violeta Magdalena Rojas Huayta, Rebecca Pradeilles, Edwige Landais, Nervo Verdezoto, Emma Haycraft, Michelle Holdsworth.

**Writing – review & editing:** Violeta Magdalena Rojas Huayta, Rebecca Pradeilles, Hilary M. Creed-Kanashiro, Emily Rousham, Doris Delgado, Edwige Landais, Nervo Verdezoto, Emma Haycraft, Michelle Holdsworth.

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
