## [Decision Letter · Decision Letter 0]

8 Aug 2023

PONE-D-23-05420Identifying priority double-duty actions to tackle double burden of malnutrition in infants and young children in Peru: Assessment and prioritisation of government actions by national expertsPLOS ONE

Dear Dr. Rojas Huayta,

Thank you for submitting your manuscript to PLOS ONE. After careful consideration, we feel that it has merit but does not fully meet PLOS ONE’s publication criteria as it currently stands. Therefore, we invite you to submit a revised version of the manuscript that addresses the points raised during the review process.

We look forward to receiving your revised manuscript.

Kind regards,

Chandan Kumar, Ph.D.

Academic Editor

PLOS ONE

Journal Requirements:

2. In the ethics statement in the Methods, you have specified that verbal consent was obtained. Please provide additional details regarding how this consent was documented and witnessed, and state whether this was approved by the IRB

Funding: This study was supported by the UK Medical Research Council (MR/S024921/1) and CONCYTEC/FONDECYT Perú (032-2019) through the Newton Fund. 

5. Please remove your figures from within your manuscript file, leaving only the individual TIFF/EPS image files, uploaded separately. These will be automatically included in the reviewers’ PDF.

Reviewers' comments:

Reviewer's Responses to Questions

**Comments to the Author**

1. Is the manuscript technically sound, and do the data support the conclusions?

Reviewer #1: Partly

Reviewer #2: Yes

2. Has the statistical analysis been performed appropriately and rigorously? 

Reviewer #1: N/A

Reviewer #2: N/A

3. Have the authors made all data underlying the findings in their manuscript fully available?

Reviewer #1: Yes

Reviewer #2: Yes

4. Is the manuscript presented in an intelligible fashion and written in standard English?

Reviewer #1: Yes

Reviewer #2: Yes

5. Review Comments to the Author

Reviewer #1: This paper addresses an incredibly important issue, and understanding the potential for nutrition policy actions which can address all forms of the double burden of malnutrition is a pressing priority. The paper has a great research question and overall a good research approach to answering the question. However, I found the presentation of the results of this analysis to just be too complex and confusing to get a cohesive picture of what is going on in Peru. The authors need to find a way to summarise their findings more effectively, perhaps by collapsing categories or summarising sections, or perhaps by using tables or figures to make the results more digestible. This applies particularly in the Discussion section where I hoped to find some more synthesis.

It would also help in interpreting this information to know what the nutrition policy goals and objectives of the Peru government are to understand more the context of all the multiple policies discussed in the paper. Specific comments below.

Introduction

- This section needs to include more information on the Peru context in two ways.

o Firstly what are considered drivers of DBM in this setting, some more background on the nutrition situation in Peru could be included.

o What actually is the overarching government policy towards nutrition in the country. What are the stated aims? Is the double burden actually a priority? Are policy developments aimed at this double burden?

Methods

- I think that this section needs to be restructured or reordered. I was quite confused until around half way through what you had actually done.

- For example you introduce your study participants at the start of the methods and say they will be ‘validating the evidence document’ without us knowing what that is.

- I think you could be more clear and step by step in describing your study design.

Discussion

- Although we have a ‘Summary of findings’ section I don’t find that this section achieves its aim. There is so much complexity in this paper in indicators, domains, levels of implementation, levels of evidence regarding implementation etc. The authors really need to find an effective way to summarise all this information and give a clear picture about how overall the Peru government is going in achieving impact on DBM. In reading the results and discussion it was really not clear what the government was doing well and what they were not doing well, or what they intended to do well but had barriers to implementing. I am not sure how this can be done with so much material produced by the study, but it needs to be attempted.

- As I mentioned in the introduction section you need some information on what Peru government policies are for nutrition. You could return to this in the Discussion and reflect on policy achievements against this, as well as against all of the indicators you have used.

- Also to broaden the impact of the paper an assessment of what the Peru government has done well and what lessons for the international community more broadly can be drawn from this work would be helpful.

Reviewer #2: This manuscript reports a review of government actions to solve double burden of malnutrition in infants and young children in Peru. The study is well designed. The paper is well written with detailed information and includes discussion of different viewpoints on double duty actions. However, there are some points that should be addressed. My comments are as follows:

The tool created to assess the DDA in this study is a useful innovation. The authors should discuss how other researchers/countries should modify/adjust it to best suit their situation.

- Some indicators especially the specific micronutrient supplementation may not be applicable to other settings/countries. For example, vitamin A deficiency may not be a problem but iodine deficiency is. So these indicators should be relevant to the country-specific problems/issues.

- It might also be helpful to include maternal nutrition during lactation as one indicator.

- Nutrition-specific interventions has a limit in reducing DBM prevalence. To further reduce it, nutrition-sensitive interventions by non-health ministries will play a more important role. For infrastructure support indicators, a platform for interaction between health and non-health governmental organizations such as agriculture, trade, education will be useful.

Experts point out that there is a serious shortage of health workers, especially in counselling work. The authors should discuss how to deal with this problem. For example, using digital educational resources to reduce the burden on staff and educate caregivers and families should be considered.

6. PLOS authors have the option to publish the peer review history of their article (what does this mean?). If published, this will include your full peer review and any attached files.

Reviewer #1: No

Reviewer #2: No

---

## [Author Response · Author response to Decision Letter 0]

2 Nov 2023

Reviewer #1

1.This paper addresses an incredibly important issue, and understanding the potential for nutrition policy actions which can address all forms of the double burden of malnutrition is a pressing priority. The paper has a great research question and overall a good research approach to answering the question.

Authors’ responses

Thank you for your positive feedback.

Reviewer #1

2. However, I found the presentation of the results of this analysis to just be too complex and confusing to get a cohesive picture of what is going on in Peru. The authors need to find a way to summarise their findings more effectively, perhaps by collapsing categories or summarising sections, or perhaps by using tables or figures to make the results more digestible. This applies particularly in the Discussion section where I hoped to find some more synthesis.

Authors’ responses

We have reflected on your comment. We have added a summary of the main findings of this study, which are summarised in two parts at the beginning of the Discussion section: the policy measures that Peru is implementing well and those that need to be strengthened. This summary of findings allows the reader to get a coherent picture of what is happening in Peru.

See lines 485-528 on page 22-23

Reviewer #1

3. It would also help in interpreting this information to know what the nutrition policy goals and objectives of the Peru government are to understand more the context of all the multiple policies discussed in the paper.

Authors’ responses

Thanks for the suggestion. In the introduction, the goals and objectives of the Peruvian government's nutrition policy have been added to better understand the context of all the multiple policies discussed in the document.

This has been added lines 79 to 90 on page 4

Reviewer #1

4. Introduction

- This section needs to include more information on the Peru context in two ways.

o Firstly what are considered drivers of DBM in this setting, some more background on the nutrition situation in Peru could be included.

o What actually is the overarching government policy towards nutrition in the country. What are the stated aims? Is the double burden actually a priority? Are policy developments aimed at this double burden?

Authors’ responses

We have added background information on the situation of the double burden of malnutrition in Peru, as well as: the country's governmental nutrition policy and stated objectives; the existence of policies aimed at addressing the double burden of malnutrition; and the prioritisation of the double burden of malnutrition in national policies.

This has been added lines 61 to 90 on page 3-4

Reviewer #1

5. Methods

- I think that this section needs to be restructured or reordered. I was quite confused until around half way through what you had actually done.

- For example you introduce your study participants at the start of the methods and say they will be ‘validating the evidence document’ without us knowing what that is.

- I think you could be more clear and step by step in describing your study design.

Authors’ responses

We have adopted your suggestion and have subsequently reordered the section to help simplify the process for readers.

See lines 94 to 243 on page 4-11

Reviewer #1

6. Discussion

- Although we have a ‘Summary of findings’ section I don’t find that this section achieves its aim. There is so much complexity in this paper in indicators, domains, levels of implementation, levels of evidence regarding implementation etc. The authors really need to find an effective way to summarise all this information and give a clear picture about how overall the Peru government is going in achieving impact on DBM. In reading the results and discussion it was really not clear what the government was doing well and what they were not doing well, or what they intended to do well but had barriers to implementing. I am not sure how this can be done with so much material produced by the study, but it needs to be attempted. 

Authors’ responses

Thanks for the suggestion, we have improved the wording of the "summary of findings" section to enhance understanding of what the Peruvian government is doing well and what needs to be strengthened to tackle the double burden of malnutrition in children under 2 years of age.

See lines 485-528 on page 22-23

Reviewer #1

7.- As I mentioned in the introduction section you need some information on what Peru government policies are for nutrition. You could return to this in the Discussion and reflect on policy achievements against this, as well as against all of the indicators you have used.

Authors’ responses

Your suggestion has been adopted and information on the Peruvian government's policies on nutrition has been added to the introduction and discussion section.

See lines 79 to 90 on page 4

See lines 497-512 on page 22

See lines 523-528 on page 23

Reviewer #1

8.- Also to broaden the impact of the paper an assessment of what the Peru government has done well and what lessons for the international community more broadly can be drawn from this work would be helpful.

Authors’ responses

Thank you, your suggestion was included in the discussion section

See lines 497-512 on page 22-23

Reviewer #2

1. This manuscript reports a review of government actions to solve double burden of malnutrition in infants and young children in Peru. The study is well designed. The paper is well written with detailed information and includes discussion of different viewpoints on double duty actions. However, there are some points that should be addressed.

Authors’ responses

Thank you for your overall positive feedback

Reviewer #2

2. The tool created to assess the DDA in this study is a useful innovation. The authors should discuss how other researchers/countries should modify/adjust it to best suit their situation.

Authors’ responses

Thank you for the suggestion, which is now included in the section on strengths and limitations.

See lines 672-682 on page 28-29

Reviewer #2

3. Some indicators especially the specific micronutrient supplementation may not be applicable to other settings/countries. For example, vitamin A deficiency may not be a problem but iodine deficiency is. So these indicators should be relevant to the country-specific problems/issues.

- It might also be helpful to include maternal nutrition during lactation as one indicator.

Authors’ responses

Thank you for the suggestion, which is now included in the section on strengths and limitations.

See lines 678-682 on page 30

Reviewer #2

4. Nutrition-specific interventions has a limit in reducing DBM prevalence. To further reduce it, nutrition-sensitive interventions by non-health ministries will play a more important role. For infrastructure support indicators, a platform for interaction between health and non-health governmental organizations such as agriculture, trade, education will be useful.

Authors’ responses

We agree with your point of view. Among the infrastructure support indicators, there is the intragovernmental interaction platform indicator, which was evaluated in this study. We have added more detail on this indicator in the results and discussion section.

See lines 274-278 on page 12

See lines 504-509 on page 22-23

Reviewer #2

5. Experts point out that there is a serious shortage of health workers, especially in counselling work. The authors should discuss how to deal with this problem. For example, using digital educational resources to reduce the burden on staff and educate caregivers and families should be considered.

Authors’ responses

Thank you for the suggestion, we agree, we have added a paragraph about this in the discussion section

See lines 662- 666 on page 29

---

## [Decision Letter · Decision Letter 1]

6 Mar 2024

PONE-D-23-05420R1Identifying priority double-duty actions to tackle double burden of malnutrition in infants and young children in Peru: Assessment and prioritisation of government actions by national expertsPLOS ONE

Dear Dr. Rojas Huayta,

Thank you for submitting your manuscript to PLOS ONE. After careful consideration, we feel that it has merit but does not fully meet PLOS ONE’s publication criteria as it currently stands. Therefore, we invite you to submit a revised version of the manuscript that addresses the points raised during the review process.

We look forward to receiving your revised manuscript.

Kind regards,

Rohullah Roien

Academic Editor

PLOS ONE

Reviewers' comments:

Reviewer's Responses to Questions

**Comments to the Author**

1. If the authors have adequately addressed your comments raised in a previous round of review and you feel that this manuscript is now acceptable for publication, you may indicate that here to bypass the “Comments to the Author” section, enter your conflict of interest statement in the “Confidential to Editor” section, and submit your "Accept" recommendation.

Reviewer #2: All comments have been addressed

2. Is the manuscript technically sound, and do the data support the conclusions?

Reviewer #2: Yes

3. Has the statistical analysis been performed appropriately and rigorously? 

Reviewer #2: Yes

4. Have the authors made all data underlying the findings in their manuscript fully available?

Reviewer #2: Yes

5. Is the manuscript presented in an intelligible fashion and written in standard English?

Reviewer #2: No

6. Review Comments to the Author

Reviewer #2: The final version of the manuscript should be re-checked for typological errors and English corrections. For example on line 77, it should read ... 39.8% of children aged 6-59 months suffered from anaemia.

7. PLOS authors have the option to publish the peer review history of their article (what does this mean?). If published, this will include your full peer review and any attached files.

Reviewer #2: No

---

## [Author Response · Author response to Decision Letter 1]

13 Mar 2024

1. If the authors have adequately addressed your comments raised in a previous round of review and you feel that this manuscript is now acceptable for publication, you may indicate that here to bypass the “Comments to the Author” section, enter your conflict of interest statement in the “Confidential to Editor” section, and submit your "Accept" recommendation.

Reviewer #2: All comments have been addressed

Authors’ responses

Thank you for your overall positive feedback

2. Is the manuscript technically sound, and do the data support the conclusions?

Reviewer #2: Yes

Authors’ responses

Thank you for your overall positive feedback

3. Has the statistical analysis been performed appropriately and rigorously?

Reviewer #2: Yes

Authors’ responses

Thank you for your overall positive feedback

4. Have the authors made all data underlying the findings in their manuscript fully available?

Reviewer #2: Yes

Authors’ responses

Thank you for your overall positive feedback

5. Is the manuscript presented in an intelligible fashion and written in standard English?

Reviewer #2: No

Authors’ responses

Thank you for your comment, we have checked the typographical and grammatical errors.

6. Review Comments to the Author

Reviewer #2: The final version of the manuscript should be re-checked for typological errors and English corrections. For example on line 77, it should read ... 39.8% of children aged 6-59 months suffered from anaemia.

Authors’ responses

Thank you for your comment. We have again reviewed the final version of the manuscript for typological errors and corrections in English. We have also corrected the error noted in line 77

---

## [Decision Letter · Decision Letter 2]

29 Apr 2024

Identifying priority double-duty actions to tackle double burden of malnutrition in infants and young children in Peru: Assessment and prioritisation of government actions by national experts

PONE-D-23-05420R2

Dear Dr. Rojas Huayta,

We’re pleased to inform you that your manuscript has been judged scientifically suitable for publication and will be formally accepted for publication once it meets all outstanding technical requirements.

Kind regards,

Charles Odilichukwu R. Okpala

Academic Editor

PLOS ONE

Additional Editor Comments (optional):

Acceptable for publication.

Reviewers' comments:

Reviewer's Responses to Questions

**Comments to the Author**

1. If the authors have adequately addressed your comments raised in a previous round of review and you feel that this manuscript is now acceptable for publication, you may indicate that here to bypass the “Comments to the Author” section, enter your conflict of interest statement in the “Confidential to Editor” section, and submit your "Accept" recommendation.

Reviewer #2: All comments have been addressed

Reviewer #3: All comments have been addressed

2. Is the manuscript technically sound, and do the data support the conclusions?

Reviewer #2: Yes

Reviewer #3: Yes

3. Has the statistical analysis been performed appropriately and rigorously? 

Reviewer #2: Yes

Reviewer #3: Yes

4. Have the authors made all data underlying the findings in their manuscript fully available?

Reviewer #2: Yes

Reviewer #3: Yes

5. Is the manuscript presented in an intelligible fashion and written in standard English?

Reviewer #2: Yes

Reviewer #3: Yes

6. Review Comments to the Author

Reviewer #2: (No Response)

Reviewer #3: (No Response)

7. PLOS authors have the option to publish the peer review history of their article (what does this mean?). If published, this will include your full peer review and any attached files.

Reviewer #2: No

Reviewer #3: **Yes: **Agatha Aduro-Agema

---

## [Editor Report · Acceptance letter]

8 May 2024

PONE-D-23-05420R2 

PLOS ONE

Dear Dr. Rojas Huayta, 

I'm pleased to inform you that your manuscript has been deemed suitable for publication in PLOS ONE. Congratulations! Your manuscript is now being handed over to our production team.

Kind regards, 

on behalf of

Dr. Charles Odilichukwu R. Okpala 

Academic Editor

PLOS ONE